# Primary care networks as a means of supporting primary care: findings from qualitative case study-based evaluation in the English NHS

Kath Checkland ,[1] Donna Bramwell,[1] Lynsey Warwick-Giles,[1] Simon Bailey ,[2] Jonathan Hammond [1]

¹School of Health Sciences, The University of Manchester, Manchester, UK
²Centre for Health Services Studies, University of Kent, Canterbury, UK

**Correspondence to**
Dr Kath Checkland;
Katherine.H.Checkland@
manchester.ac.uk

## ABSTRACT

**Objectives** This study aimed to evaluate primary care networks (PCNs) in the English National Health Service. We ask: How are PCNs constituted to meet their defined goals? What factors can be discerned as affecting their ability to deliver benefits to the community, the network as a whole and individual members? What outcomes or outputs are associated with PCNs so far? We draw policy lessons for PCN design and oversight, and consider the utility of the chosen evaluative framework.

**Design and setting** Qualitative case studies in seven PCN in England, chosen for maximum variety around geography, rurality and population deprivation. Study took place between May 2019 and December 2022.

**Participants** PCN members, staff employed in additional roles and local managers. Ninety-one semistructured interviews and approximately 87 hours of observations were undertaken remotely. Interview transcripts and observational field notes were analysed together using a framework approach. Initial codes were derived from our evaluation framework, with inductive coding of new concepts during the analysis.

**Results** PCNs have been successfully established across England, with considerable variation in structure and operation. Progress is variable, with a number of factors affecting this. Good managerial support was helpful for PCN development. The requirement to work together to meet the specific threat of the global pandemic did, in many cases, generate a virtuous cycle by which the experience of working together built trust and legitimacy. The internal dynamics of networks require attention. Pre-existing strong relationships provided a significant advantage. While policy cannot legislate to create such relationships, awareness of their presence/absence is important.

**Conclusions** Networked approaches to service delivery are popular in many health systems. Our use of an explicit evaluation framework supports the extrapolation of our findings to networks elsewhere. We found the framework to be useful in structuring our study but suggest some modifications for future use.

## STRENGTHS AND LIMITATIONS OF THIS STUDY

⇒ The use of an explicit evaluation framework provides a structured approach to the assessment of the impact of the primary care networks policy in England, including wider implications for the establishment of such networks elsewhere.

⇒ The study has also tested and refined the framework, enhancing its usefulness for future studies.

⇒ Data from interviews were triangulated with rich and nuanced data from 87 hours of meeting observations.

⇒ The study explored the development of primary care networks over a 3-year period, allowing some assessment of their development over time; however, the policy has continued to evolve, and so the results represent a snapshot in time.

⇒ It is too early to assess the impact of primary care networks quantitatively; our assessment, therefore, focuses on the implementation of the policy and qualitative assessment of achievements.

## INTRODUCTION

As health systems across the world wrestle with the need to provide co-ordinated care to an ageing population, the development of networks of healthcare organisations has come to be seen as an important mechanism by which such care can be delivered safely and effectively.[1 2] This organisational form has been particularly attractive in health systems which have adopted an approach characterised by the contracting out of service provision to a variety of service providers.[3 4] The benefits of networked organisational forms are said to include: better co-ordination; enhanced problem-solving capacity; better services for clients; and greater resilience.[2 5 6] In the National Health Service (NHS) in England, the latest manifestation of this trend is seen in the development of primary care networks (PCNs).[7] Underpinned by an add-on contract to the standard general medical services contract which governs the provision of primary care services in England, general practices are incentivised to work together



in groups covering populations of 30–50 000 people to provide a range of additional services.[8] [9] PCNs are seen as an important means by which collaborative services will be delivered across what are being called 'neighbourhoods' (ie, locally coherent geographical footprints covering populations about the size of local government electoral wards[10]), supporting general practices to work more closely with other community services, increasing practice resilience and enabling the development of a coherent primary care 'voice' within the system.[11]

In this paper, we explore the extent to which these policy aspirations are likely to be met. In doing this, we provide evidence to both inform future policy in England and to support those in other systems considering the development of similar networked forms of primary care provision. We use a framework for assessing network effectiveness developed by Cunningham *et al*.[1] Building on their experiences of evaluating the impact of a variety of health system networks, these authors synthesised decades of interdisciplinary research on networked organisational forms to develop an evaluative framework which was tested and refined in a number of stakeholder workshops. We use this framework to structure the findings from an evaluation of the early stages of PCN development, considering the factors likely to affect their effectiveness and providing early evidence as to how the desired policy outcomes may or may not be achieved in both the short and long term. Using evidence from an evaluation of the first few years of PCN operation, we highlight areas in which the policy framework underpinning PCNs could usefully be developed or adjusted, and provide some thoughts on the usefulness of the evaluation framework adopted.

## Evaluating networks

Cunningham *et al*[1] draw attention to the complexities of researching healthcare networks, highlighting the fact that, as is often the case in social science disciplines, the term 'network' does not have a stable or unique meaning. In particular, they suggest that the term 'is often used as a synonym for 'partnership', 'collaboration', 'alliance' and 'group'' (Cunningham[1], p2). In generating their evaluative framework, they define networks as: 'the structure of relationships between people, groups or organisations, joined together through nodes and ties' (Cunningham[1], p2). From this perspective, networks can be spontaneous or mandated, but all share the important characteristic of specific and definable ties between network members. Using this definition, partnerships or collaborations may be networks as long as they include durable relationships, recognisable ties and collective goals which transcend the goals of the individual organisations.[12] Thus, for example, a project-specific collaborative group assembled to deliver a specific programme (such as, the temporary collaborations assembled to deliver the Vanguard integrated care pilots in the English NHS[13]) is a collaboration but not a network, as it involves collective goals but not durable ties. However, if the group were to continue beyond the length

of the programme and collectively pursue new areas of work they could plausibly be described as a network. Such definitions are neither undisputed nor unchallengeable. For example, some authors would be less concerned with the notion of collective goals, including in their definition social networks for which collective goals would be hard to identify.[14] To ensure clarity, our focus here is on networked forms of organisation which are often identified as being a form of governance distinguishable from either hierarchies (in which a central authority mandates action) or markets (in which self-interested organisations compete and pursue temporary alliances to achieve their goals).[12] From this perspective, networked organisations offer a mechanism by which service provision can be orchestrated rather than directly provided ('steering, not rowing'[15]), allowing greater control than is possible in a pure market while outsourcing the practicalities of service delivery to the networked organisations. PCNs clearly fall into this definition, although 'in the shadow of hierarchy'[16] in the form of centrally mandated targets and incentives.[17]

In their review of the literature on networks, Cunningham *et al*[1] draw extensively from the organisational studies and public policy literatures. Their evaluation framework is set out in figure 1.

They start by highlighting the fact that if we are to understand the factors affecting the effectiveness of any given network we must first have a clear idea as to its nature and mode of operation. This includes: defining the goals of the network (either intrinsic or, for mandated networks, extrinsically defined); understanding the details of how network members relate to one another, manage themselves and share resources; and understanding the context within which any given network exists. Such an understanding is important because it has been shown that such factors are relevant in determining the extent of network effectiveness.[18]

Second, they identify the fact that the impact of networks must be considered across a range of scales. In publicly funded systems, the community is a legitimate stakeholder, including the population served by the network, but also other healthcare providers in the local area, regulators, local politicians and relevant consumer groups. Beyond this, the perspective of the network as a whole must be considered. This includes the network itself (ie, the organisation (however loose) which network members join) and any orchestrating entity, such as health system hierarchy or regional administrative body responsible for oversight. In spontaneous networks such an entity will not be present, but in the majority of networked organisations delivering services on behalf of health systems there will be some sort of national or regional oversight, ensuring quality at the very least and often controlling resources. Finally, they argue that the perspective of individual members of the network are of importance. Thus, for PCN members, the benefits or disbenefits of belonging to a network must be considered. It is theoretically possible that an organisational network

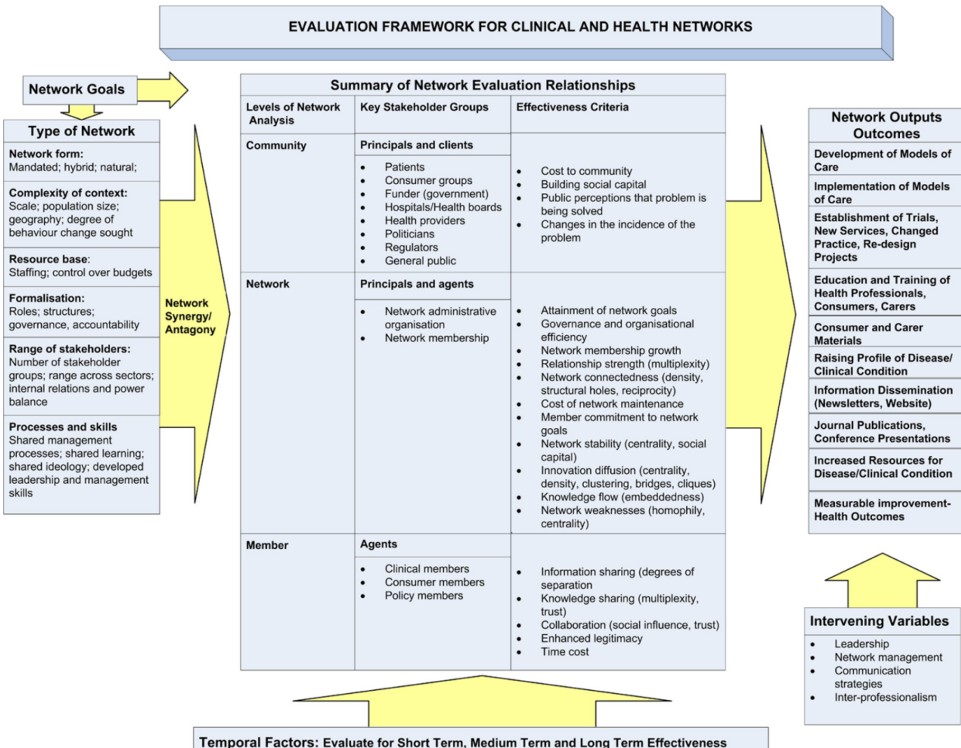

**Figure 1** Cunningham framework for network evaluation. Adapted from Cunningham *et al.*[1]

could generate significant benefits overall while still disadvantaging individual members. This might occur, for example, if a network were to redistribute resources from well-resourced members to those less well resourced. The extent to which the overall (network-level) benefits outweigh these local disbenefits would depend on the values and normative framework within which the network is operating.

They next identify what they call 'effectiveness criteria' at each of these levels: community, network and member. At community level, they highlight such things as social capital, public perception of improvements and reduced incidence of particular problems; at network level, they identify a large number of potential criteria, including characteristics of network functioning such as stability, cohesiveness and relationship strength; while at member level they suggest that criteria should include successful knowledge sharing and increased trust between member organisations. In operationalising this framework it became clear to us that these 'criteria' in fact represent intermediary mechanisms which evidence suggests underpin the effectiveness of a network, rather than criteria by which effectiveness might be judged. Thus, for example, a cohesive and stable network might have a better chance than a fragmented one of achieving its goals, but the achievement of those goals would be influenced by other factors such as resource availability, effective management processes and a supportive context. In our version of the framework, we have, therefore, labelled these important factors as 'mechanisms with potential to increase effectiveness', and consider the evaluative activity

to be exploring the extent to which these intermediate mechanisms are or are not present.

The framework finally encourages the evaluator to identify desired outcomes and outputs associated with the relevant network. Many of these will flow directly from the declared goals of the network, but others may be emergent. For example, a network established to co-ordinate and streamline care for a particular clinical condition might find that successful networking also facilitated research activity or public engagement. Outputs might include procedures, guidelines or new service provision, while outcomes are more distal and refer to demonstrable improvements in relevant metrics. The authors identify 'intervening variables' which might be expected to affect the translation of network activity into outcomes, including leadership, management, communication and interprofessionalism. However, it could also be argued that these things are design features of the network, and so should be considered during the initial network characterisation.

### PCN in England

PCNs were established in 2019 as a result of negotiations between the BMA (representing general practitioner (GP) practices) and NHS England (the arms' length body responsible for overseeing the management of the NHS, initially known as the NHS Commissioning Board[19]). Essentially, practices were offered the opportunity to obtain additional investment and support in return for working together as networks to deliver additional services over and above those provided under the

**Table 1** PCN funding as of the 2020/2021 network specification (from NHSE figures)

| Funding stream | Money available (yearly) | Basis of payment | Proportion of available contract funding | Weighted |
|---|---|---|---|---|
| Network participation payment | £1.761 per weighted patient registered with practice | Prospective weighted capitation payment | 13% | Carr-Hill formula |
| Additional roles reimbursement scheme | £7.131 per weighted patient registered with PCN practices | Weighted reimbursement for 100% of salary and employer costs for additional roles (up to maximum/role) | 52.6% | Carr-Hill Formula |
| PCN support payment | £0.27 per weighted patient registered with PCN practices. (1 April 2020–30 September 2020 COVID-19 payment) | Prospective weighted capitation payment—transferred from the Investment and Impact Fund due to COVID-19 pandemic | 2% | Carr-Hill formula |
| Core PCN funding | £1.50 per patient registered with PCN practices | Prospective unweighted capitation payment | 11% | No weighting |
| Clinical Director contribution | £0.722 per patient registered with PCN practices | Prospective unweighted capitation payment for 0.25WTE/50 000 patients | 5.3% | No weighting |
| Extended hours access | £1.45 per patient registered with PCN practices | Prospective unweighted activity-based payment for 30 minutes/1000 patient/week | 10.7% | No weighting |
| Care home premium | £60 per care home bed (rising to £120 from 1 April 2021) | Prospective unweighted capitation payment per care home bed linked to the PCN | Variable | No weighting |
| Investment and impact fund | £111 per point (initially 194 points available per PCN starting 1 October 2020) | Activity-based payment dependent on points gained adjusted for prevalence and list size | 5.3% | No weighting—revalence and list size adjustments |

NHSE, National Health Service England; PCN, primary care network.

standard General Medical Services contract. As such they represent an attempt by NHS England to orchestrate the provision of improved services to local populations and clearly fall into our definition of a network. They were not mandatory, but as the investment associated with PCNs represented the majority of additional investment available for general practices, nearly 100% joined when the contract went live.[20] Engagement with a PCN provided a variety of additional income streams, some directed at individual practices and some at the network as a whole.[21] Table 1 sets out these sources of funding.

Payments include: a payment for participating; funding to support the employment across the network of additional clinical and other staff (known as the Additional Roles Reimbursement Scheme, ARRS); payment for a clinical director; payment for providing additional services, including routine appointments outside normal working hours and additional care for patients living in care homes; a support payment to provide some infrastructure for the network; and access to an incentive scheme (known as the investment and impact fund, IIF) by which additional funds will accrue for meeting a series of targets.[17] The criteria by which these payments are made have been modified due to the COVID-19 pandemic, with some incentive or service-related payments distributed without the need for the services to be provided or targets met in order to support the pandemic response. Each year from the inception of PCNs, we have observed an increased proportion of general practice funding being provided through the PCN mechanism, as opposed to direct to the practice. The extent that an individual PCN and practice relate to each other financially is variable, depending on factors such as the intra-network financial agreements and the extent the entities engage with pay for performance mechanisms (such as the quality and outcomes framework, the IIF and other enhanced services). NHS digital states it provided a mean of £163.65 per patient in total for all general practice, including

PCNs in 2021/2022. Unfortunately, missing data makes analysis of the PCN funding streams challenging. We estimate that on 2021/2022 the maximum reimbursable amount a PCN may receive, prior to the IIF, is £26.49 per patient, which thus represents a significant proportion of available funding for each practice. However, the actual reimbursed figure may be less. The range of additional services that it is intended that PCNs will eventually provide is wide, including: structured medication reviews for a defined population of patients, anticipatory care planning, additional services for care homes, early cancer diagnosis, cardiovascular risk management and tackling neighbourhood health inequalities.[17] The COVID-19 pandemic delayed the implementation of some of these, but during this time PCNs have been engaged collectively in the delivery of the COVID-19 vaccination programme. Practices working in a network are required to have in place a network agreement,[22] and each one is required to have a designated leader known as a clinical director. No funds were initially provided for managerial support, but additional funds have been provided for this purpose in the third year of their operation.[23]

Early study of PCNs revealed that they were established to fulfil a number of policy objectives, including: as a means of supporting primary care resilience and stabilising primary care; to work with other providers across a defined geographical footprint to deliver a wider range of services; and to provide a collective 'voice' for primary care within a reorganised system.[24] These objectives are not necessarily straightforwardly related to one another: a network optimised to support its constituent practices might look quite different from one optimised to work with external providers. An initial evaluation found that, while practices had engaged with the policy and formed themselves into networks, there was significant variability in size, configuration and the maturity of relationships, with those who had worked together previously at some advantage. Meso-level support from a commissioning authority was felt to be important but sometimes lacking, and managerial support for network functions and development was seen as important[25]

In this paper, we use the framework developed by Cunningham et al[1] to present the findings from a 3-year evaluation of PCN development, addressing the following research questions:

► How are PCNs constituted to meet their defined goals?
► What factors can be discerned as affecting their ability to deliver benefits to the community, the network as a whole, and individual members?
► What outcomes or outputs are associated with PCNs so far?

In answering these questions, we draw together policy lessons for PCN design and oversight more generally, and consider the utility of the chosen evaluative framework.

## METHODS

We undertook a three-phase qualitative study of PCN establishment and development. An initial phase (reported elsewhere[24]) used interviews with policy makers and documentary analysis to identify the policy goals underlying PCN development. Phase 2 comprised telephone interviews with staff working in 37 clinical commissioning groups (CCGs), interviewees were responsible for supporting local PCN development.[26] The findings from this phase informed the third qualitative phase, the development of longitudinal qualitative case studies in seven PCNs in six CCG areas (July 2020–March 2022). Case study sites were selected to reflect heterogeneity. PCN size, population demographics, PCN structure and geography were accounted for in our sampling strategy (see table 2 PCN characteristics).

Ninety-one respondents took part in semistructured interviews and approximately 87 hours of observations were undertaken remotely across the case study sites by authors DB, LW-G, JH and SB, all of whom were outsiders in the contexts in question. Recruitment continued in each case study sites until saturation with respect to our evaluation questions was reached. All participants gave informed consent. The initial topic guide was derived from our evaluation framework and is included as online supplemental file. Topics included: early experiences of establishing the PCN; the factors affecting this; progress in tackling network goals; experiences of employing staff across networks; outcomes which might be attributed to the PCN; and the impact of COVID-19. Interviews and observations were undertaken via Microsoft Teams and Zoom due to restrictions that were in place because of the COVID-19 global pandemic. Interviewees were chosen to represent the full range of people involved with PCN development and operation, including PCN clinical directors, additional roles PCN staff, managers, GPs and people from the PCN membership, local commissioners and staff from local provider organisations. Topics covered in the interviews included individuals' roles and experiences in the development of the PCN. Observations were undertaken of PCN meetings and wider meetings that PCNs were involved in, for example, community programmes of work. These observations were undertaken to understand the work PCNs were undertaking alongside exploring the governance structures and accountability arrangements that had been established in practice. Furthermore, additional documentation was collected from PCNs, where they were obtainable. Documentation included PCN network agreements, meeting agendas and minutes. Data collection continued until the research team agreed that we had a good understanding of each case study site in context. Short summaries of each site were produced, bringing together evidence from documents, interviews and observations, and interpretations checked with key informants where there was any discrepancy between sources or where information was lacking.

All of the data were coded and analysed by the research team, using NVivo software (V.12). A framework analysis

| Table 2 | Case study PCN characteristics | | | | | | |
|---|---|---|---|---|---|---|---|
| | PCN A | PCN B2 | PCN B3 | PCN C | PCN D | PCN E1 | PCN E2 |
| GP member practices | 10–15 | 15–20 | 10–15 | 5–10 | 5–10 | 5–10 | 5–10 |
| Patient population | 60 000–70 000 | 90 000–100 000 | 70 000–80 000 | 50 000–60 000 | 80 000–90 000 | 30 000–40 000 | 30 000–40 000 |
| Population deprivation | High | Mixed | High | Mixed | Mixed | High | High |
| Rurality (approx. %) | 0.5 | 1.5 | 0.1 | 0.1 | >20 | 2 | 1 |
| Structure | Flat Practice model | Flat practice model | Flat Practice model | Flat Practice model | Partnership model | Limited Company | Limited Company |
| Collaboration history | Mixed collaborative history (some new practices added to the existing GP practice group to form the PCN) | Mixed collaborative history (some practices had worked closely together) | Limited previous collaboration | Loose collaboration links through previous CCG initiatives | Strong collaborative history | Practical collaborative history through previous national initiatives | Practical collaborative history through previous national initiatives |

CCG, clinical commissioning group; GP, general practitioner; PCN, primary care network.

approach was employed.[27] The coding framework was developed iteratively, with some codes developed prior to data collection based on existing literature and the chosen evaluation framework. Other codes were introduced and developed throughout the project based on the data that had been gathered.

### Patient and public involvement

Members of the public were involved in an advisory group which supported the development and conduct of the study.

## RESULTS

Our previous work has explicated the multiple and potentially conflicting policy goals associated with PCNs.[24] In this section, we explore the realities on the ground as PCNs have developed, and consider the impact on their potential to meet these differing goals.

### The structure and make up of PCNs

Within broad guidance, PCNs were encouraged to develop in ways that suited their local environment. An initial requirement for a population coverage of between 30 and 50 000 people[7] was flexed in practice and established PCNs varied considerably in size, with 35% of PCNs covering populations larger than 50 000.[20] Bringing together this wider evidence with the findings from our case studies, the nature of PCNs as networks can be summarised using the categories specified by Cunningham et al[1] (see table 3).

### Factors affecting the potential effectiveness of PCNs in meeting their goals
#### Community level

One of the key policy goals associated with PCNs in England is to support the development of neighbourhood-based collaborative service provision, working with other providers across their geographical area, with an assumption that this type of integrated working would lead to more care being provided outside hospitals. In practice, we found that in our case study sites this was not generally a priority, in part because incentives associated with PCNs are focused on internal activity, but also because practices felt themselves to be under pressure and therefore unable to engage with additional activity beyond their core service delivery. Within this, there were a number of factors which seemed to influence the extent to which PCNs were able to engage more widely.

First, the extent of pre-existing working relationships in a local area were important. In many sites, pre-existing collaborative arrangements at neighbourhood level were operational before PCNs were announced. These 'neighbourhood teams' were often orchestrated in some way by the local commissioning organisation or driven by local collaborations such as federations, and often included representatives from GP practices, the community trust,

**Table 3** Network characteristics—structure and make up

| Characteristic | Manifestation in PCNs |
|---|---|
| Network form | ► PCNs are technically voluntary. However, the fact that PCNs represent the majority of additional investment available for primary care means that incentives for participation are strong, although the threat of leaving the scheme does give practices some leverage with respect to their regulatory authority. PCN goals, arrangements and activities are mandated to a significant degree by government. |
| Context: scale, population size, geography | ► PCNs vary greatly in size, with a range in coverage from around 15 000 to more than 200 000 people<br>► PCNs vary in configuration, with a mean of 5 practices involved, but 34 PCN include a single practice, while 77 include more than 10 practices<br>► PCNs vary in internal context, with some made up of more or less equal sized practices, while others are heterogeneous, with some dominated by a single large practice<br>► PCNs vary in geography, with those in urban areas more likely to have overlapping geographical footprints |
| Resource base: staffing, finance, buildings, etc | ► Management is highly variable, with some employing dedicated managers and others not.<br>► Financial resources available to PCNs are complex and tied to contractual obligations. More than 50% of the money available is tied up in the employment of additional staff, and a further 15%–20% tied to the delivery of specific services or meeting particular targets<br>► The availability of office space for new staff is highly variable and quite limited in some PCNs |
| Formal structures: roles, structures, governance, accountabilities | ► PCNs had considerable leeway in establishing their internal structures, with a standard interpractice agreement[22] setting out only the bare essentials such as means of joining or leaving the network. The extent to which formal structures have been established varies, and there are particular issues around the employment and payment of additional staff in some areas<br>► Internal accountabilities are not yet formally established in many PCNs, with some confusion as to how far clinical directors and other PCN leaders can be held to account, by whom for what<br>► External accountability is to NHS England, but the operation of this accountability is also not fully established. |
| Range of stakeholders: inside and outside networks | ► Some PCN goals focus clearly on engaging with a range of stakeholders, including other NHS service providers and voluntary groups, but there is considerable variation in the extent to which this has yet occurred<br>► There has been minimal engagement with patients or the wider community in most PCNs<br>► Many PCNs have some sort of local inter-PCN body or group, including federations, local 'networks of networks' or, in some places, not for profit organisations. |
| Processes and skills: shared management processes, available management/leadership skills | ► The extent to which PCNs have the management and leadership process and capabilities that they need is very variable and still developing |

NHS, National Health Service; PCN, primary care network.

the mental health service provider and third sector organisations.

The extent to which pre-existing neighbourhood teams and newly constituted PCNs were able to integrate and work together varied between sites. In site A, where neighbourhood teams were well established and resourced, the arrival of PCNs was associated with attempts to neighbourhood teams 'wrap neighbourhood teams around PCNs' (N03032, October 2020). One interviewee talked about the PCN being the 'yolk' in the neighbourhood team 'egg', and emphasised the necessity of a functioning general practice collaborative entity to the productive operation of the neighbourhood team (N720sr, October 2020). However, PCNs are constituted on the basis of practice populations, not geography and this could cause confusion:

…so the ['neighbourhood teams'] would have been based on a geographic footprint of working together. The PCNs have been on the basis of, well, we own this practice, this practice, this practice and this practice. And that's caused no end of confusion for some people in terms of how that then kind of comes together. [N3701q]

Furthermore, the contractual requirements of PCNs through the GP contract were perceived as a barrier to more extensive integration of PCNs into neighbourhood teams.

So in some areas, the PCNs are working really well in partnership. And in other areas, I think they're just not as advanced in their ways of working with other providers round the table. And I think, to be honest,

it's probably been confusing for some of them because ['neighbourhood teams'] came first. […] Well, neither has precedence, you just work together, that's the whole purpose of what we're doing. You know, nobody…it just so happens the PCNs have got the money.

And that has, to be honest, probably caused some friction because actually when that money gets kind of split out, it doesn't encourage the PCNs to work in partnership, it encourages them to work within their own footprint. [N3701q, July 2021 _Site]

In Site C, a pre-existing neighbourhood team model was in place and PCN arrangements mapped closely to the footprints of these teams. Efforts were underway at the site level to adapt the provision of community services so that it was more coherent with the PCN and neighbourhood team geography. The nature of the dynamic, however, between PCNs and neighbourhood teams was somewhat unclear with different interviewees framing this differently. For example, one interview said this:

Where do PCNs stop and where do neighbourhoods start. The reality is there will be some things that PCNs will lead on in terms of projects and in terms of delivery, there will be some things that neighbourhoods lead on and there will be some things that we will do collectively together. I think partners, how do I say this, what we've, primary care networks I think creates a perception that they are led by and owned by primary care and you're putting primary care right at the centre and obviously you've got clinical leadership that is there. But actually in terms of delivering population health needs, that isn't just primary care that's delivering on that on their own, it needs to be them alongside their other partners as well. […] So primary care is a really, really important building block, but it has to be partnered as equal working together. [N46026-Site C]

Whereas another talked about it in this way:

Yeah, so Neighbourhood and Networks, I keep saying to the Neighbourhoods, and we're doing a lot of work around that, we need to stop referring to them as Neighbourhoods and Networks, 'cause they're all as one, essentially. So, we call them Networks within Neighbourhoods, all these pilots are taking place within the Networks, on behalf of the wider Neighbourhood if you like. […] Neighbourhoods preceded Networks, but actually, and probably unknowingly at the time, actually they were delivering on the Primary Care Network model. [N520KK-Site C]

These different perceptions of the role of PCNs in Neighbourhood working need to be resolved if cross-sector working is to be effectively established.

Beyond the general question of neighbourhood-working. PCNs were required to participate in multidisciplinary team working to provide additional services to people living in Care Homes. Under the compulsory 'Enhanced Care in Care Homes Service Specification', PCNs were required to allocate Care Home Residents to a participating practice and to work with other agencies to deliver comprehensive care.[28] In some areas such services had already been established, but in others this was not the case. The specifics of the funding and contractual arrangements in place could render this problematic, as this quote from a representative of a community service provider organisation makes clear:

So that's where some of the arguments came around care homes and enhanced care in care homes because primary care has now been incentivised to do that through that enhanced care in care homes scheme. […]

What we're saying to primary care is, yes, of course we'll do what we can, but we haven't been given any additional resource and actually we're trying to do all the other work that would fall by the wayside if we did that. So that's quite challenging. But yes, if we really want community and primary care to work in an integrated way, we need the same incentives and the same contract. [N1018c, Nov 21, Site A]

In a small number of our sites more ambitious programmes of cross-sector working were being established, but this was very resource intensive, requiring clinical directors to work well-beyond their contracted hours to develop the relationships required to support this type of activity. Adequate and flexible management support was an important factor in allowing this to take place.

Finally, policy guidance suggests that one of the advantages of PCNs is that they will be able to work across their neighbourhoods to established wider programmes of Population Health Management, identifying and targeting support towards high risk individuals in a neighbourhood. In practice we did not find this to be a priority for our case study PCNs, with limited understanding among those we interviewed as to what this approach might involve.

Overall, we found that community-level neighbourhood working, while acknowledged as a potential benefit of PCNs, has yet to be established. Those areas with pre-existing good relationships with other providers were at an advantage, but current incentives and available resources do not fully support this activity. Developing the required trusting local relationships is time-consuming, and requires managers and clinical leaders with the time and skills to invest in this activity.

### Network level

At the network level, the key policy objective was around supporting general practices to realise the claimed benefits of at-scale working and to stabilise a care sector under significant pressure. The main mechanism by which this is to be achieved is via funding for additional staff—known as the Additional Roles Reimbursement Scheme,

**Table 4**  ARRS staff roles over time

| Initial roles 2019/2020—target recruitment 20 000 staff | Additional roles 2021/2024—target recruitment 26 000 staff |
|---|---|
| Clinical pharmacists (2019) | Pharmacy technicians |
| Social prescribing link workers (2019) | Care co-ordinators |
| Physiotherapists—first contact | Health coaches |
| Physician associates | Dieticians |
| Paramedic (April 2021) | Podiatrists |
| | Occupational therapists |
| | Mental health professionals |
| | Nurse associates |
| | Nurse training associates |
| | Advanced practitioners |
| | General practice assistant (from 2022) |
| | Digital and transformation lead (from 2022) |
| | Adult mental health practitioner (from 2022) |
| | Children and Young People Mental Health Practitioner (from 2022) |

ARRS, Additional Roles Reimbursement Scheme.

ARRS—and via the softer benefits of working together. The ARRS provides direct reimbursement of 100% for staff employed to work across the PCN. Table 4 sets out the types of staff who could be recruited and the numbers available across all PCNs.

PCNs are not themselves legal entities, and so are unable to employ staff. This means that a variety of contractual mechanisms can be used to provide staff under the ARRS scheme, including:

► Employment by a single 'lead practice' on behalf of the PCN.
► Employment by another legal entity such as a legally constituted Federation or other body.
► Contracting for services from another entity such as an NHS Community Trust or a voluntary sector service provider for the provision of a service—may not always be the same individual.
► Contracted from an agency, such that the workers are independent contractors.

Some of our case study PCNs suggested that they would prefer not to be the 'lead practice' employing staff because of the implications when there are disciplinary issues, or pensions administration. One suggested that the reluctance of some practices to employ staff directly was underpinned by scepticism over the longevity of PCNs, which led to concerns about liability for redundancy payments should the scheme cease. In one PCN the practices will

have nothing to do with the employment of ARRS staff. Our findings mirror those of Baird et al,[29] who studied the early employment of ARRS staff and highlighted the crucial importance of managerial and HR support.

Recruitment to these new roles tended to be more straightforward in those PCNs with established trusting relationships, but in many areas there have been problems in filling some roles, associated sometimes with a shortage of particular professionals.

> … pharmacists play one PCN off against the other to get a higher banding. So that it's created that intra—PCN war of attrition, in some places, that people, well, just play each other off to get the highest banding that they possibly can. Which you can't blame staff for doing, but it's creating probably some inequities, maybe, within the system. Some people will pay higher and take the risk that they've got a shortfall that they'll make the money up with. Others will say, no, we won't go higher than this specific banding. N0303t_231020_LWG

Contracting with other organisations for staff did not always alleviate these issues, and ARRS funds were not always spent.

> 'In the other one, ARRS fund has not been utilised. It's been contracted, there are two pharmacists for 17 practices of 90,000 patients . There's a physiotherapy service that we have again contracted through another private company, but they don't have enough staff to provide to all the 17 practices so currently they are only providing it to seven practices and they are advertising to recruit more physiotherapists but there aren't any new recruits that they were able to successfully make. So kind of literally we haven't utilised our ARRS fund.' N570mu_090721_DB

Funding for ARRS staff is relatively rigid, with a number of rules in place which could sometimes make things difficult. For example, funds unspent in 1 year cannot be carried forward into the next, and in the first 2 years the types of staff who could be employed were highly specified. Funds unused could not be recycled into other aspects of patient care, and only salaries could be funded. Most of our case study sites were also struggling with accommodation for their new staff, as many buildings were full without spare clinical spaces. There is no additional funding in the scheme for estates, and so this was potentially problematic:

> We're not that lucky, we've got nowhere to go. Then there was a bit of…to me, a bit of an issue around funding. So we've been forced to make networks in, what, 2019 this all started. We're getting forced to become a network. We're getting forced to recruit staff and spend money. No one's given us anywhere to live. So we've got staff on the streets basically. [N011c6_B_Feb22]

Our participants also told us that integrating the new staff into practices could be difficult, particularly if their roles were not understood or if they were moving between practices and only spending short amounts of time in each. Training, development and supervision were all required, and this was not always straightforward for newer roles:

So that now all comes into it, which is a good thing, it's not a bad thing, because all of these staff need development. But, the training pathway and the support packages are coming out now and they're too late. For example, the trainee nurse associate package, you know, people have already got them in place, and they've got different pathways, and now you're saying that they have to go on this accredited one, so it's going backwards to go, and I appreciate you sometimes need to go backwards to go forwards, but it's unnecessary. [N601jg_SB]

In summary, the ARRS was welcomed, but has proved quite complicated to operationalise, with complexities around many aspects of the scheme. It is too soon to tell whether the anticipated alleviation of GP workloads has materialised, with some respondents telling us that the supervision required for new staff meant that time savings may be limited. PCNs with trusting relationships were at an advantage in employing staff, and in the longer term the experience of working together to employ staff collaboratively in this way may be beneficial in cementing those relationships.

### Member level

The COVID-19 pandemic provided an early test of the extent to which PCNs were able to provide the softer benefits associated with working more closely together at scale that were anticipated in the policy. PCNs were deeply involved in the pandemic response, supporting practices by establishing local 'hot' hubs to assess patients with COVID-19 and participating in the vaccination programme. Many respondents told us that the experience in the pandemic had acted to accelerate the development of trusting relationships, but it was also true that where interpractice relationships were poor or dysfunctional this could prevent effective collaborative working.

It brought us together in a way that crises can do, so it brought us together as an organisation, and particularly we moved quickly to harmonise quite a few of our processes to get us through COVID, and there's been a lot of cross-site working in cross cover and help, so it's really brought us together from that point of view. [N060fj_D_Apr21]

Again, I don't really know where it all changed and why it changed. I know that we've…I feel a lot closer to people since we've started doing COVID clinics. 'Cause I'm very much involved in the COVID clinics and I've got a really good relationship, and we never had a relationship with [X] before. And I've got

a really good relationship with both the doctor and the practice manager there. I don't know where it all changed, I don't know if it was when we were starting to talk about clinics, and it was such a massive relief knowing that [X] took on the bulk of that work… [N250wt_A_Mar21]

Initially, it would…I don't know how to say it really without sounding derogatory, but it was like every man for himself….So, the big practices were like, right we're sorted bish-bosh, while little practices were, oh my God help me, what about me. And it got quite…we were having lots and lots of Teams meetings trying to sort it all out.[N050oz_A_Mar21]

The latter quote highlights one of the important factors determining how PCNs have been able to work together: their internal configuration and relative size/power of constituent practices. As noted above, PCNs are heterogeneous, with some made up of practices of similar sizes and covering similar populations, while others are much more mixed, for example one very large practice and a number of smaller ones.[20] In the longer term, these dynamics are likely to be important, as practices within PCNs are required to work together to deliver services and collaborate to meet incentive targets. Good internal relationships, with mutual trust and reciprocity, will be very important in this. Our study suggests that it remains early days for PCNs, and these dynamics will take time to settle down.

### Outputs/outcomes

It remains early days to judge how far PCNs and the associated funding have delivered beneficial changes for practices, populations and the wider system. ARRS staff have been recruited, and are delivering services for patients, although, as discussed, the scheme is yet to fully mature.[29] PCNs were successful in delivering services in the pandemic, particularly collaborating to provide 'hot hub' sites to care for patients with COVID-19 and delivering a significant proportion of the vaccination programme.[30] Many of the services to be delivered were delayed by the pandemic, but enhanced services for care homes are up and running in most areas, and PCNs are currently developing plans to tackle neighbourhood health inequalities and to provide anticipatory care. We found that working together to deliver specific services can have beneficial feedback effects of developing trust and improved relationships. This, in turn, suggests that it is important that PCNs are allowed time to develop, as the intended beneficial effects are likely to be contingent on the successful development of these relationships.

### DISCUSSION

We have used Cunningham *et al*'s[1] evaluation framework to structure an exploration of the development of PCNs in England. We found that PCNs have been successfully established across England, but that progress is variable,

with a number of factors important in determining how far groups of practices have been able to develop strong working relationships. While the specific operation of these factors is particular to the UK context, the evaluation framework allows us to extrapolate to consider the more general lessons for the delivery of services via networks in other contexts.

First, we found that good managerial support was helpful in many aspects of PCN development, from operationalising the ARRS to supporting the development of collaborative relationships both within the PCN and with other organisations. The initial funding of PCNs did not include dedicated funding for managerial support, and this meant that the extent of management available to our case study sites was highly variable, with consequent variability in the effectiveness of many aspects of PCN operation. More recently an additional stream of funding has been introduced to support a management function, and our study suggests that this will be important in their ongoing ability to meet policy goals.

Second, we found that the requirement to work together to meet the specific threat associated with the global pandemic did, in many cases, generate a virtuous cycle by which the experience of delivering services collectively built trust and legitimacy, thereby enhancing their ability to work together in the future. This suggests that when establishing new networks, governing authorities could usefully consider incentivising an early requirement to deliver a specific service and create the conditions for reduced friction in the realisation of such a service. Being rewarded for working together in this way is likely to support the development of collaborative relationships, while the experience of working together will quickly highlight potential problematic areas or issues which need attention.

Third, the internal dynamics of networks are important, and require attention. In particular, decision-making processes, mechanisms for undertaking shared activities and rules around resource allocation and sharing can have a significant impact on network ability to function. The relative size and power of constituent units will be important here and must be considered when configuring rules and working practices. These things are, in turn, affected by the specific requirements of contracts, incentives and available resources, and these require careful design as well as the provision of support from a local governing authority to mediate disputes, support decision-making and provide guidance and oversight.

Finally, pre-existing strong relationships were important, with those groups which had worked together successfully in the past at a significant advantage. Clearly policy cannot legislate to create such trusting relationships, but awareness of the potential impact is important. Local governing authorities could provide additional support and resources to those groups without pre-existing relationships, and funding mechanisms could ensure that networks have adequate internal management support to do the work required to build structures and collaborative working processes.

More generally, the incentive payments associated with PCN engagement were clearly important in ensuring that practices engaged with the process, at least initially. Indeed, although participation was not compulsory, NHS England reported that more than 99% of GP practices have joined a PCN.[8 31] This suggests that the amount of funding associated with PCNs was enough to drive initial engagement. However, as we have documented, not all available funding could be used due to shortages of staff to recruit, and incentive schemes are yet to be fully operational. Further research will be required to tease out in detail the impact of the different types of funding (eg, incentive payments vs payments for staff) on PCN activity and engagement.

The principal strength of our study lies in our use of an explicit evaluation framework to guide data collection and analysis. This provides transparency and allows the explicit comparison of our findings to evaluations of networked organisational forms in primary care elsewhere. In addition, our interview data were triangulated with rich and nuanced data from ethnographic observation of PCN meetings, increasing the credibility of our findings. While we were able to track developments over a period of 2–3 years, this nevertheless represents a snapshot in time of the operation of a 5-year contract. This means that it was not possible to reliably assess outcomes quantitatively, and further longer-term evaluation would be valuable. In the context of the global COVID-19 pandemic, both interview and ethnographic data collection were achieved online. In keeping with other studies, remote interviews were found to be an effective and efficient way of carrying out interviews,[32] particularly with busy clinical staff whom we may have struggled to speak to face to face. However, remote observation of meetings was more limiting, with less opportunity to observe interactions between those not speaking and limited opportunities to build rapport with participants.

There is a considerable and eclectic literature on networked organisational forms, spanning a number of different disciplines, from organisational studies to implementation science. The Cunningham *et al*'s framework[1] was developed following review of these literatures, alongside input from a range of experts. We found it to be a useful approach in structuring our research questions and in organising a significant volume of data. We found particular value in its focus on community, network and member levels. This provided a structure within which to consider the different potential beneficiaries of network activity, and highlighted the fact that different design features will be relevant in considering benefits for different groups. However, we struggled to operationalise the factors labelled as 'effectiveness criteria' within the central element of the framework, and eventually came to regard these as 'mechanisms with the potential to increase effectiveness' rather than as criteria by which effectiveness could be judged. Moreover, we

found that 'intervening variables' which the framework suggests will affect outcomes could rather be thought of as features of design which could be modified. Finally, we would argue for a more explicit acknowledgement within the framework of the role and responsibilities of the national or regional governing authority. Where networked approaches to service design are mandated, they will include a macrolevel/national authority setting rules and specifying available resources, and a mesolevel/regional authority with responsibility for supporting the operationalisation of the networked approach. Their role is important in all aspects of network operation and must be considered in any evaluation.

**Acknowledgements** We are grateful to our participants for giving us their time and allowing us to observe their activities. We are also grateful to the wider project team for stimulating discussions and engagement with the project findings.

**Contributors** KC conceived the project and developed the initial design. She collaborated on the data analysis, wrote the first draft of the manuscript and finalised it for publication. She is accountable for all aspects of the work adn is the guarantor. DB led data collection in some sites and contributed to the qualitative analysis. She contributed to the intellectual development of the findings presented in the paper, edited manuscript drafts and has agreed the final submission. She is accountable for all aspects of the work. LW-G led data collection in some sites and contributed to the qualitative analysis. She contributed to the intellectual development of the findings presented in the paper, edited manuscript drafts and has agreed the final submission. She is accountable for all aspects of the work. SB led data collection in some sites and contributed to the qualitative analysis. He contributed to the intellectual development of the findings presented in the paper, edited manuscript drafts and has agreed the final submission. He is accountable for all aspects of the work. JH contributed to the project design and finalised the protocol. He led data collection in some sites and contributed to the qualitative analysis. He contributed to the intellectual development of the findings presented in the paper, edited manuscript drafts and has agreed the final submission. He is accountable for all aspects of the work.

**Funding** This research is funded by the National Institute for Health Research (NIHR) Policy Research Programme, conducted through the Policy Research Unit in Health and Social Care Systems and Commissioning, PR-PRU-1217-20801.

**Disclaimer** The views expressed are those of the author(s) and not necessarily those of the NIHR or the Department of Health and Social Care.

**Competing interests** The authors have declared that their institutions received grant funding to support this project from the National Institute of Health Research Policy Research Programme.

**Patient and public involvement** Patients and/or the public were involved in the design, or conduct, or reporting, or dissemination plans of this research. Refer to the Methods section for further details.

**Patient consent for publication** Not applicable.

**Ethics approval** This study involves human participants and was approved by University of Manchester Research Ethics Committee (Ref: 2019-6922-11177). Participants gave informed consent to participate in the study before taking part.

**Data availability statement** No data are available. Data underlying this article are not available in order to preserve the anonymity of the participants.

**ORCID iDs**
Kath Checkland http://orcid.org/0000-0002-9961-5317
Simon Bailey http://orcid.org/0000-0001-9142-2791
Jonathan Hammond http://orcid.org/0000-0002-4682-9514

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
