## [Reviewer comments · BMJ Open]

ARTICLE DETAILS

TITLE (PROVISIONAL)	Primary Care Networks as a means of supporting primary care: findings from qualitative case study-based evaluation in the English NHS
AUTHORS	Checkland, Kath; Bramwell, Donna; Warwick-Giles, Lynsey; Bailey, Simon; Hammond, Jonathan

VERSION 1 – REVIEW

REVIEWER	Hughes, Gemma University of Oxford, Nuffield Department of Primary Care Health Sciences
REVIEW RETURNED	20-Jul-2023

GENERAL COMMENTS	This is an important and timely study of the development of primary care networks, which considers carefully the way in which policy changes interrelated with previous local arrangements and organisational relationships. I have two minor suggestions for revisions: 1. I am interested to know a little more about the investment in PCNs in relation to the research questions of this study (namely investment as a factor that affects PCNs ability to deliver benefits). For example, on page 6 line 16 – the additional investment associated with PCNs is noted, and the relative proposition of this investment in terms of other additional funding – however it would be helpful to quantify this (perhaps simply in terms of the overall investment by NHSE). Table 1 provides the cost per head available to practices – it would be useful to understand what this meant in terms of the proportion of practice funding. The point is clearly made that the additional investment made PCNs, if not mandatory, hard to reject by practices but I think readers would be interested to understand this in monetary terms, or as e.g. a proportion of average practice income. On page 8, lines 24-31, the authors note that practices found it hard to engage in additional activity beyond their core service delivery and on page 11 difficulties in utilising the funds for ARRRS. Whilst a fuller analysis of the investment and value created would be beyond the scope of the paper (though interesting to know if this is planned?) a brief additional section in the discussion section of the paper reflecting on the extent to which the investment helped or hindered PCNs would, I think, add to this paper.2. The only other area that I would like a little clarification/commentary is in relation to methods (page 7). The authors note that the research was carried out remotely via Teams or Zoom – it would be helpful to have a brief commentary on any limitations/strengths of conducting this research remotely – for example, did the remote nature of observations limit what was
--

	possible to observe? At line 43, the authors state that data collection continued until they agreed they had a good understanding of the sites – was there any participant/member checking to validate this agreement at all?
REVIEWER	Sitienei, Jackline University of the Witwatersrand Centre for Health Policy, Public Health
REVIEW RETURNED	25-Jul-2023
GENERAL COMMENTS	Effort put in this paper is visible except for minor revision on the of the structure of the abstract.

VERSION 1 – AUTHOR RESPONSE

Reviewer 1:

We are grateful for the positive assessment of our paper and constructive suggestions for improvement.

Investment in PCNs

The reviewer requested that we contextualise our discussion of the amount of funding available to PCNs via reference to the overall funding of GP practices. We agree that this is a useful addition, and have added text setting out the total amount of funding available per weighted patient via PCNs and comparing this with the total funding provided to GP practices on page 5. Whilst not all available funding was used, the resulting figure of PCN funding equal to approximately 7.5% of total funding demonstrates its scale. We have also added text to the discussion (p13) discussing the impact of these payments and highlighting the need for further research on this topic.

The reviewer also requested some reflection on our use of remote interviewing and observations. We have added discussion of this to the 'strengths and limitations' section of the discussion (p13), highlighting the value of remote interviewing but acknowledging the limitations of remote observation of meetings.

Editor:

*Please update the SRQR checklist submitted with your manuscript to indicate where within the manuscript each reporting requirement is addressed (eg, by subsection). – Checklist done and page numbers added

*In the article title, 'evaluation study' is not very informative about the study design. Please revise to clarify, ensuring consistency with the abstract and main text (eg, qualitative case study-based evaluation?). -Changed

*Please update the abstract 'Objectives' section to avoid use of bullet points. Each section in the abstract should be a single paragraph of text. --Changed

*Please update the 'Design and setting' section of the abstract to include the dates between which the study took place. - Added

*Please update the 'Participants' section of the abstract to indicate the number of participants and the study methods (how were data collected from these participants [interviews and meeting observations?], and how were the data analysed?). - Added

*Please revise the fourth bullet point in the 'Strengths and Limitations' section so that it consists of a single sentence (eg, via use of a semi-colon)._ Changed

*In the main text, you indicate that "Ninety-one semi structured interviews" were undertaken, but it is not stated if this means there were 91 interviewed individuals (each interviewed once). Please revise to clarify. Please also clarify how this sample size was determined.- Changed

*Was a topic guide used for the semi-structured interviews? Please include a copy of this as a supplemental file, cited in the main text where the topics covered are described. – Changed and uploaded

*The sentence 'Members of the public were involved in an advisory group which supported the development and conduct of the study' should be placed under the heading 'Patient and public involvement' at the end of the main text Methods section (immediately before the Results section). - Changed

*Please ensure that you have fully discussed the methodological limitations of the study in the Discussion section of the main text, including full discussion of any key limitations highlighted in the 'Strengths and Limitations' section after the abstract. Changed

*Please update the 'Conflict of interest statement' to clarify if there are any other COIs. If not, please add "The authors declare that they have no other competing interests."- Done

*Please add a statement at the end of the manuscript under the heading 'Data availability statement'. This should indicate whether and how the data underlying the study will be available for sharing with other researchers.- Done

VERSION 2 – REVIEW

REVIEWER	Hughes, Gemma University of Oxford, Nuffield Department of Primary Care Health Sciences
REVIEW RETURNED	22-Sep-2023
GENERAL COMMENTS	The authors have responded to previous review comments. This is an important and timely paper.

VERSION 2 – AUTHOR RESPONSE